# A Comprehensive Review of the Effects of Different Simulated Environmental Conditions and Hybridization Processes on the Mechanical Behavior of Different FRP Bars

**Mohammadamin Mirdarsoltany** [1,*] **, Farid Abed** [2] **, Reza Homayoonmehr** [1] **and Seyed Vahid Alavi Nezhad Khalil Abad** [3]

[1] Department of Civil Engineering, Amirkabir University of Technology, Tehran 15875-4413, Iran; rhomayoonmehr@aut.ac.ir
[2] Department of Civil Engineering, American University of Sharjah, Sharjah P.O. Box 26666, United Arab Emirates; fabed@aus.edu
[3] Department of Civil Engineering, Birjand University of Technology, Birjand 97175-569, Iran; svalavi@birjandut.ac.ir
[*] Correspondence: amin.st@aut.ac.ir

**Abstract:** When it comes to sustainability, steel rebar corrosion has always been a big issue, especially when they are exposed to harsh environmental conditions, such as marine and coastal environments. Moreover, the steel industry is to blame for being one of the largest producers of carbon in the world. To supplant this material, utilizing fiber-reinforced polymer (FRP) and hybrid FRP bars as a reinforcement in concrete elements is proposed because of their appropriate mechanical behavior, such as their durability, high tensile strength, high-temperature resistance, and lightweight-to-strength ratio. This method not only improves the long performance of reinforced concrete (RC) elements but also plays an important role in achieving sustainability, thus reducing the maintenance costs of concrete structures. On the other hand, FRP bars do not show ductility under tensile force. This negative aspect of FRP bars causes a sudden failure in RC structures, acting as a stumbling block to the widespread use of these bars in RC elements. This research, at first, discusses the effects of different environmental solutions, such as alkaline, seawater, acid, salt, and tap water on the tensile and bonding behavior of different fiber-reinforced polymer (FRP) bars, ranging from glass fiber-reinforced polymer (GFRP) bars, and basalt fiber-reinforced polymer (BFRP) bars, to carbon fiber-reinforced polymer (CFRP) bars, and aramid fiber-reinforced polymer (AFRP) bars. Furthermore, the influence of the hybridization process on the ductility, tensile, and elastic modulus of FRP bars is explored. The study showed that the hybridization process improves the tensile strength of FRP bars by up to 224% and decreases their elastic modulus by up to 73%. Finally, future directions on FRP and hybrid FRP bars are recommended.

**Keywords:** composite bars; FRP bars; hybrid FRP bars; GFRP bars; BFRP bars; CFRP bars; durability; hybridization process; alkaline solution; seawater solution

## 1. Introduction

Population growth is causing an increased demand for the development of infrastructures and a tremendous consumption of materials. Two of the most important materials for infrastructures are concrete and steel [1]. The corrosion of steel bars is a major problem that causes a heavy cost for repairing or replacing concrete elements. Thus, it is not feasible to use steel bars as a reinforcement in seawater sea-sand concrete because of the corrosion [2–8] of these types of bars [9–14], as well as the aggressive environmental conditions that exist near to coastal areas [15–24]. Seawater and sea sand contain chloride ions. These ions speed up corrosion development and negatively affect the long performance of concrete structures [25–29]. The corrosion of steel bars causes the cracking, staining, and

spalling of concrete cover [26,30]. It also reduces the cross-sections of bars, thus, shortening the service life of reinforced concrete (RC) structures [31–33]. Furthermore, in comparison with FRP bars, steel bars emit more $CO_2$ into the environment [34]. This is contrary to the durability and sustainability of reinforced concrete structures, which are a popular topic nowadays, as such non-durable structures can cause vast economic and environmental problems, especially for infrastructures [5,35,36]. There are several methods to address the steel corrosion problem in reinforced concrete structures, such as using a hybrid system in which steel and composite bars are used simultaneously in the reinforced concrete section, enhancing the properties of concrete elements to decrease their permeability [37,38], and by employing carbon nanotubes to reduce rebar corrosion [39–43].

Another viable approach to dealing with the aforementioned issues is by using fiber-reinforced polymer (FRP) bars. This is due to their appropriate mechanical behavior, such as their tensile strength, durability, and anti-corrosion performance resistance [44–55]. Supplanting FRP bars with conventional steel bars can preserve the mechanical behavior of reinforced concrete elements [56]. There are several different FRP bars, some of the most applicable of which are glass fiber-reinforced polymer (GFRP), aramid-fiber reinforced polymer (AFRP), carbon fiber-reinforced polymer (CFRP), and basalt-fiber reinforced polymer (BFRP) [57,58] bars. However, the usage of FRP bars in reinforced concrete elements has some downsides as well. One of the major problems is the sudden failure of these types of bars under tensile force. This can give rise to the sudden collapse of concrete structures when they are subjected to dynamic loads [59]. To improve the ductility of FRP bars in tensile loads, scholars have introduced the hybridization process [47]. The concept of this innovative process is gradually failing by selecting different fibers that rupture under different ultimate strains. Only a limited number of investigations have been carried out on the effects of the different solutions on the mechanical properties of FRP bars and hybrid FRP bars [60]. The aim of this study was to summarize the current research on the mechanical behavior of FRP and hybrid FRP bars, their bonding and tensile strength in particular, under harsh environmental conditions.

## 2. Materials Properties for Fabricating FRP Bars

### 2.1. Composite Fibers

Many types of fibers are used to develop FRP bars, including glass, aramid, carbon, and basalt fibers. Composite fibers play an essential role in the mechanical properties of FRP bars. Each type of fiber has different properties. Table 1 reports the mechanical properties of different fibers [49].

**Table 1.** Typical properties of fibers for FRP composites.

| Fiber Type | Density (kg/m$^3$) | Tensile Strength (MPa) | Young Modulus (GPA) | Ultimate Tensile Strain (%) | Thermal Expansion Coefficient (10$^{-6}$/°C) | Poisson's Coefficient |
|---|---|---|---|---|---|---|
| E-glass | 2500 | 3450 | 72.4 | 2.4 | 5 | 0.22 |
| S-glass | 2500 | 4580 | 85.5 | 3.3 | 2.9 | 0.22 |
| Alkali resistant glass | 2270 | 1800–3500 | 70–76 | 2.0–3.0 | - | - |
| ECR | 2620 | 3500 | 80.5 | 4.6 | 6 | 0.22 |
| Carbon (high modulus) | 1950 | 2500–4000 | 350–650 | 0.5 | −1.2–0.1 | 0.20 |
| Carbon (high strength) | 1750 | 3500 | 240 | 1.1 | −0.6–0.2 | 0.20 |
| Aramid (Kevlar 29) | 1440 | 2760 | 62 | 4.4 | −2.0 longitudinal 59 radial | 0.35 |
| Aramid (Kevlar 49) | 1440 | 3620 | 124 | 2.2 | −2.0 longitudinal 59 radial | 0.35 |
| Aramid (Kevlar 149) | 1440 | 3450 | 175 | 1.4 | −2.0 longitudinal 59 radial | 0.35 |
| Aramid (Technora H) | 1390 | 3000 | 70 | 4.4 | −6.0 longitudinal 59 radial | 0.35 |
| Aramid (SVM) | 1430 | 3800–4200 | 130 | 3.5 | - | - |
| Basalt (Albarrie) | 2800 | 4840 | 89 | 3.1 | 8 | - |

According to other research [9,61], the use of E-glass, basalt, and carbon is more common for fabricating FRP bars. Table 1 shows that the use of carbon fibers leads to higher elastic modulus and tensile strength in fabricated composite bars, compared to E-glass and basalt fibers. However, in comparison with other fibers, carbon fibers are the most expensive ones. This can be a stumbling block to the widespread use of such fibers in manufacturing composite bars.

*2.2. Resin*

Thermosetting matrices are not only used to maintain fibers and conserve the surface of the fibers, but also to help prevent the propagation of cracks in FRP bars. Furthermore, different types of resins have different extents of ability in protecting the fibers against different environmental conditions [62]. Table 2 outlines the mechanical properties of some thermosetting matrices [49].

**Table 2.** Typical properties of thermosetting matrices.

| Property | Matrix | | |
| --- | --- | --- | --- |
| | **Polyester** | **Epoxy** | **Vinyl Ester** |
| Density ($kg/m^3$) | 1200–1400 | 1200–1400 | 1150–1350 |
| Tensile strength (MPa) | 34.5–104 | 55–130 | 73–81 |
| Longitudinal modulus (GPa) | 2.1–3.45 | 2.75–4.10 | 3.0–3.5 |
| Poisson's coefficient | 0.35–0.39 | 0.38–0.40 | 0.36–0.39 |
| Thermal expansion coefficient ($10^{-6}/°C$) | 55–100 | 45–65 | 50–75 |
| Moisture content (%) | 0.15–0.60 | 0.08–0.15 | 0.14–0.30 |

With regards to Table 2, in terms of elastic modulus, epoxy resins have a higher elastic modulus in comparison with other resins. Furthermore, considering tensile strength, vinyl ester resins have higher tensile strength compared to other resins.

In addition, the types of resin play a pivotal role in the long-term performance of fabricated FRP bars. Benmokrane et al. [63] investigated the effects of different resins, namely vinyl ester, polyester, and epoxy, on the durability of GFRP bars. They found that the use of vinyl ester and epoxy for fabricating FRP bars, unlike polyester, presented the lowest degradation level in terms of physical and mechanical behavior after exposure to an alkaline solution. In other research caried out by Benmokrane et al. [64,65], the effects of vinyl ester and epoxy resin were investigated regarding the durability of BFRP and GFRP bars. They observed that glass–vinyl ester FRP bars had the best long-term performance in the alkaline solution, in comparison with basalt–epoxy and basalt–vinyl-ester FRP bars.

## 3. Different Simulated Environments and Their Effects on the Mechanical Properties of FRP Bars

The degradation of FRP bars occurs when free hydroxyl ions ($OH^-$) scatter through the matrices of FRP bars [66]. The interface between the fiber and matrix is a nonhomogeneous and thin region. This layer is also prone to deterioration. Chen et al. [66] classified this deterioration into three mechanisms, namely matrix osmotic cracking, interfacial debonding, and delamination. Moisture diffusion into FRP composites could be influenced by the material's anisotropic and heterogeneous character. In addition, wicking through the fiber–matrix interface in the fiber direction could be the predominant mechanism of moisture ingress. Nonvisible dissociation between the fibers and matrix could lead to rapid losses of interfacial shear strength. Unfortunately, limited attention has been paid to the effect of the resin system type on the physical and mechanical properties or the durability characteristics of GFRP bars. Furthermore, the types of resin play a crucial role in the degradation of FRP bars. The more ester groups there are, the more likely FRP bars are to degrade. As polyester has more ester groups than vinyl ester, with polyester being more prone to hydrolysis [50,67,68], polyester-based FRP bars are less durable than epoxy- and vinyl ester-based FRP bars. Another significant factor involved in degrading

FRP bars is the type of fiber, such as basalt, carbon, glass, and aramid fibers. Different environmental conditions have different effects on the level of degradation in the FRP bars. The combination of sodium hydroxide (NaOH), potassium hydroxide (KOH), and calcium hydroxide (Ca(OH)$_2$), with pH values of 13.6 and 12.7, leads to simulating an alkaline solution with normal- and high-strength concrete environments, respectively [69,70]. To simulate seawater, sodium chloride (NaCl) is combined with (Na$_2$SO$_4$) [66,71].

The tensile strength and bonding behavior of FRP bars that experience harsh environmental conditions can be determined according to ASTM D7205 [72] and ASTM D7913 [73], respectively.

### 3.1. Behavior of FRP Bars under Tension

Much research has been conducted to investigate the tensile behavior of FRP bars, ranging from GFRP [50] and CFRP [66], to BFRP [74,75] and AFRP [76]. The FRP bars do not exhibit any plastic behavior under tensile loading and remain linearly elastic until their ultimate strain [77,78]. Since fibers play a significant role in tensile strength, it can be stated that bars with the same diameter, appearance, and processing may have different strengths, depending on their fiber to matrix ratio [79]. The effects of harsh environmental conditions on the tensile behavior of FRP bars have been investigated by different scholars. Rifai et al. [80] tested FRP bars in alkaline solutions and concrete environments, along with different durations and temperatures. They showed that the degradation level of FRP bars was more sensitive to the conditioned temperature than to the duration.

Table 3 summarizes the tensile behavior of GFRP, CFRP, and BFRP bars, based on the bars' diameter, different environmental conditions, and time.

**Table 3.** Tensile behavior of FRP bars, based on different solutions, diameters, and time.

| Type | Bar Diameter (mm) | Solution | Days | Tensile Strength (MPa) | Ref. | Retention (%) |
|---|---|---|---|---|---|---|
| GFRP | 9.53 | Seawater | 70 | 754 | [66] | 98 |
| GFRP | 9.53 | Alkaline | 60 | 482 | [66] | 52 |
| GFRP | 6 | High-performance seawater sea sand concrete | 63 | 1036 | [81] | 97.9 |
| GFRP | 6 | Normal seawater sea sand concrete | 42 | 728 | [81] | 68.7 |
| GFRP | 19 | - | - | 633.8 | [82] | 98 |
| GFRP | 19 | - | - | 535.7 | [82] | 83 |
| GFRP | 12.7 | Saline solution | 60 | 781 | [50] | 99 |
| GFRP | 12.7 | Saline solution | 365 | 702 | [50] | 89 |
| GFRP | 6 | High-performance seawater sea sand concrete/20% | 42 | 988 | [83] | 93.7 |
| GFRP | 6 | High-performance seawater sea sand concrete/20% | 63 | 617 | [83] | 58.6 |
| GFRP | 8 | Alkaline | 45 | 1359.8 | [84] | 96.4 |
| GFRP | 8 | Alkaline | 90 | 1061.4 | [84] | 75.3 |
| GFRP | 8 | Alkaline | 135 | 994.7 | [84] | 70.5 |
| GFRP | 8 | Alkaline | 180 | 974.8 | [84] | 69.1 |
| GFRP | 8 | Seawater | 45 | 1402.6 | [84] | 99.5 |
| GFRP | 8 | Seawater | 90 | 1298.1 | [84] | 92.09 |
| GFRP | 8 | Seawater | 135 | 1275.2 | [84] | 90.4 |
| GFRP | 8 | Seawater | 180 | 1152.9 | [84] | 81.7 |
| BFRP | 6 | High-performance seawater sea sand concrete | 21 | 1341 | [81] | 99.3 |
| BFRP | 6 | Normal seawater sea sand concrete | 63 | 352 | [81] | 26 |
| BFRP | 6 | Alkaline | 21 | 1385 | [74] | 99.1 |
| BFRP | 6 | Alkaline | 63 | 852 | [74] | 60.9 |
| BFRP | 6 | Deionized water | 42 | 1320 | [74] | 94.4 |
| BFRP | 6 | Salt | 42 | 1320 | [74] | 94.4 |
| BFRP | 6 | Acid | 42 | 1301 | [74] | 93.1 |
| BFRP | 7 | Alkaline | 42 | 1012 | [75] | 60.2 |
| BFRP | 8 | Alkaline | 30 | 1409 | [75] | 89.9 |

**Table 3.** *Cont.*

| Type | Bar Diameter (mm) | Solution | Days | Tensile Strength (MPa) | Ref. | Retention (%) |
|------|------|------|------|------|------|------|
| BFRP | 6 | Alkaline | 63 | 802 | [85] | 60.58 |
| BFRP | 12 | Alkaline | 21 | 1036 | [85] | 95.18 |
| BFRP | 6 | High-performance seawater sea sand concrete/20% | 42 | 1276 | [83] | 94 |
| BFRP | 6 | High-performance seawater sea sand concrete/40% | 63 | 586 | [83] | 43.2 |
| BFRP | 8 | Alkaline | 45 | 1194.7 | [84] | 91.9 |
| BFRP | 8 | Alkaline | 90 | 1148.9 | [84] | 88.4 |
| BFRP | 8 | Alkaline | 135 | 1078.5 | [84] | 82.9 |
| BFRP | 8 | Alkaline | 180 | 1008.6 | [84] | 77.6 |
| BFRP | 8 | Seawater | 45 | 1095.2 | [84] | 84.2 |
| BFRP | 8 | Seawater | 90 | 1028.5 | [84] | 79.1 |
| BFRP | 8 | Seawater | 135 | 998.7 | [84] | 76.8 |
| BFRP | 8 | Seawater | 180 | 984.8 | [84] | 75.7 |
| CFRP | 3 | Alkaline | 70 | 2476 | [66] | 96 |
| CFRP | 8 | Alkaline | 45 | 2059 | [84] | 99.03 |
| CFRP | 8 | Alkaline | 90 | 1966.6 | [84] | 94.5 |
| CFRP | 8 | Alkaline | 135 | 1928.8 | [84] | 92.7 |
| CFRP | 8 | Alkaline | 180 | 1720.5 | [84] | 82.7 |
| CFRP | 8 | Seawater | 45 | 1894.9 | [84] | 91.1 |
| CFRP | 8 | Seawater | 90 | 1758.7 | [84] | 84.5 |
| CFRP | 8 | Seawater | 135 | 1692.5 | [84] | 81.4 |
| CFRP | 8 | Seawater | 180 | 1638.3 | [84] | 78.8 |

Figure 1 depicts the retention of different FRP bars in different environmental solutions. According to Table 3 and Figure 1, CFRP bars, regardless of the period, showed the highest resistance to the alkaline environment, compared to BFRP and GFRP bars, in terms of tensile strength. Contrary to saline and seawater solutions, GFRP bars, regardless of the duration, were more durable in comparison with other types of FRP bars.

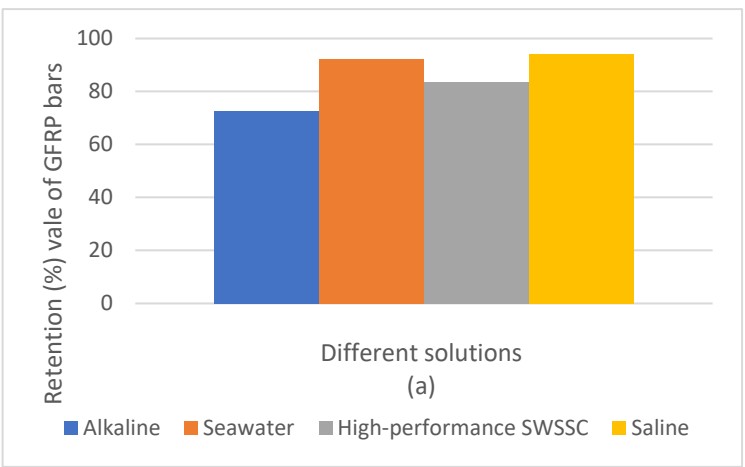

**Figure 1.** *Cont.*

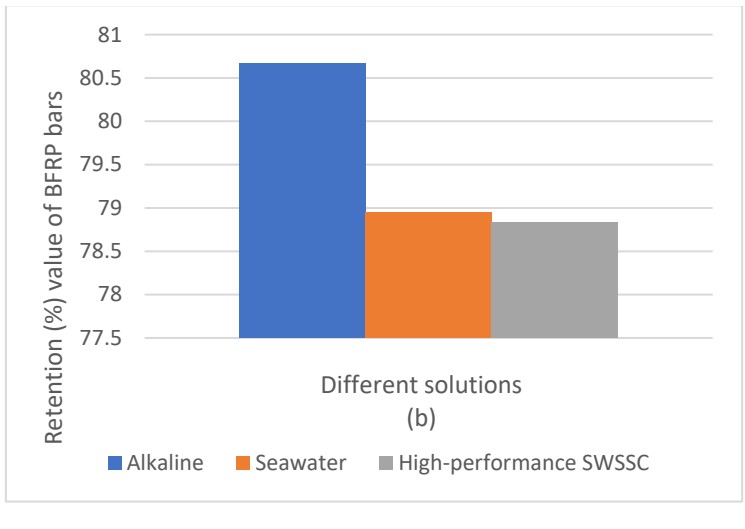

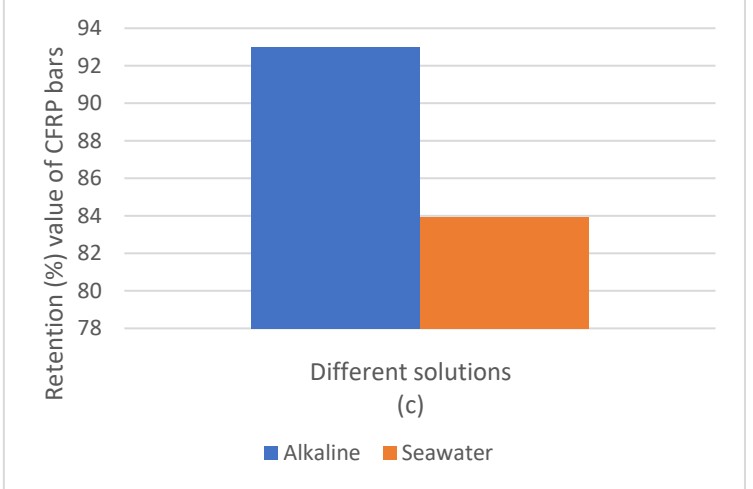

**Figure 1.** Mean retention of different FRP bars in different environmental solutions, based on data in Table 3: (**a**) retention value of GFRP bars, (**b**) retention value of BFRP bars, and (**c**) retention value of CFRP bars.

Figure 2 demonstrates the trendlines of the different FRP bars in different environmental solutions. According to Figure 2, all types of FRP bars, under different environmental conditions, experienced a downward trend. The trendlines of the BFRP bars' retention in alkaline and seawater solutions showed the lowest slope during the period. Contrary to BFRP bars in alkaline and seawater solutions, GFRP bars in seawater solutions showed the biggest downward trend during the period.

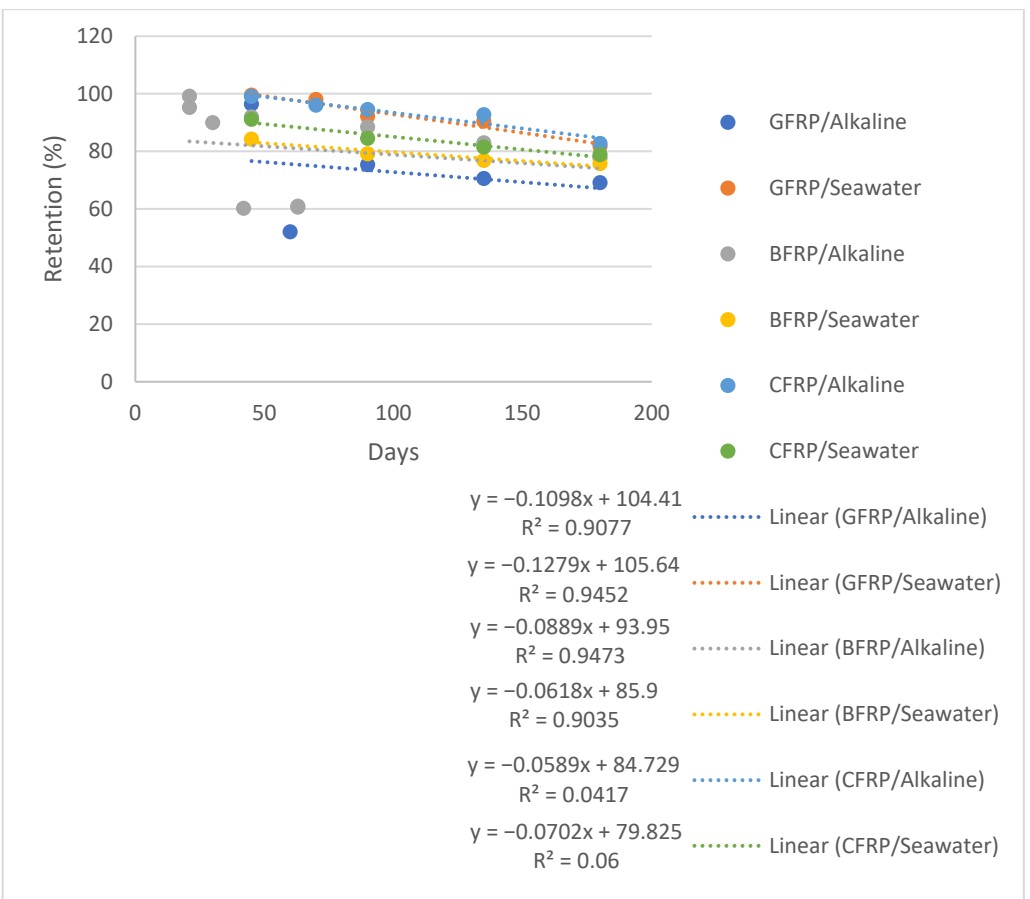

**Figure 2.** Scatter plot of tensile strength retention of different FRP bars in different environmental solutions and their trendlines, based on data in Table 3.

### 3.2. Bonding of FRP Bars to the Concrete

The mechanical properties and the durability of FRP bars depends on the long-term bond properties of the FRP bar and the concrete interface [1]. Furthermore, some critical factors play a central role in the bond behavior, including the concrete strength, concrete cover, and the concrete confinement provided by stirrup [51,86–89].

El Refai et al. [45] conducted an experiment to evaluate the bond behavior of BFRP bars compared to GFRP bars. The experiment indicated that BFRP bars have 75% of the bond behavior of GFRP bars, on average. Sharaky et al. [90] tried to assess the factors that play an essential role in the bond behavior of near-surface-mounted FRP bars, experimentally. The results showed that the capacity and the mode of failure of the specimens were affected by the adhesive properties, the FRP bar's size, and the bar surface treatment. El-Nemr et al. [71] investigated the bearing of the bond-dependent coefficient of GFRP and CFRP bars on standard- and high-strength concrete. It seems that the compressive strength of concrete would not directly or significantly affect the bonding between non-steel bars and concrete [91–93].

Apart from the bonding behavior of FRP bars under normal conditions, only a few investigations have been carried out to evaluate the effects of environmental conditions on the mechanical properties of FRP bars. El Refai et al. [94] investigated the bond strength of BFRP bars under five different environmental conditions, namely, tap water, seawater, elevated temperature, elevated temperature followed by tap water, and elevated temperature followed by seawater. The tests showed that elevated temperature, up to 80 °C, has infinitesimal effects on the bond strength of FRP bars, regardless of the fiber material. Abed et al. [95] evaluated the effects of sunlight followed by seawater and sunlight alone

on the bond strength of FRP bars. Investigations indicated that these conditions had no effect on the bonding behavior of FRP bars and concrete.

Table 4 summarizes the results of the different environmental conditions on the bond strength of FRP bars. Furthermore, Figure 3 indicates the mean value of the bond strengths of different FRP bars, under different environmental conditions.

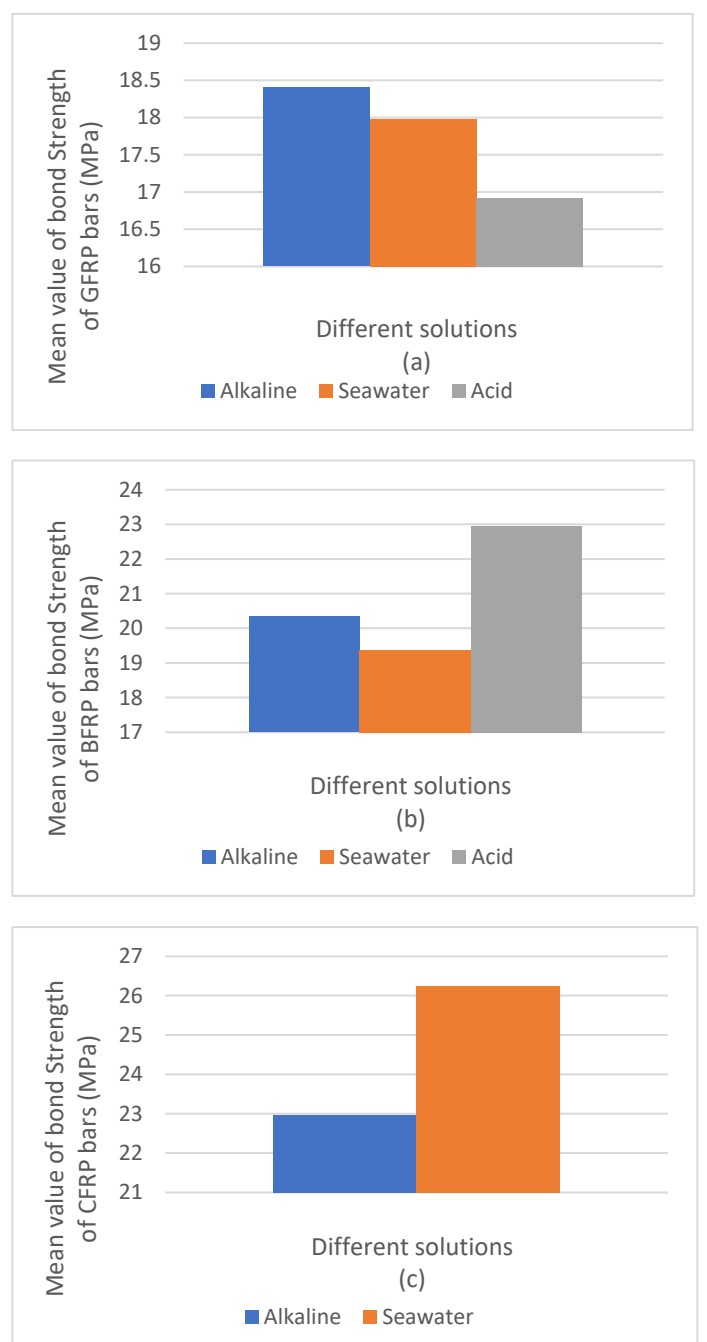

**Figure 3.** Mean retention of different FRP bars in different environmental solutions, based on data in Table 4: (**a**) retention value of GFRP bars, (**b**) retention value of BFRP bars, and (**c**) retention value of CFRP bars.

**Table 4.** Bonding behavior of FRP bars in different solutions.

| FRP Type | Bar Diameter (mm) and Sizing Shape | Solution | Temperature | Days | Mean Value of Bond Strength (MPa) | Ref. |
|---|---|---|---|---|---|---|
| GFRP | 10, Sand coating | Seawater | 23 | 60 | 14.73 | [96] |
| GFRP | 10, Sand coating | Seawater | 40 | 60 | 18.44 | [96] |
| GFRP | 10, Sand coating | Seawater | 60 | 60 | 16.29 | [96] |
| GFRP | 10, Sand coating | Seawater | 23 | 120 | 15.7 | [96] |
| GFRP | 10, Sand coating | Seawater | 40 | 120 | 14.7 | [96] |
| GFRP | 10, Sand coating | Seawater | 60 | 120 | 15.85 | [96] |
| GFRP | 10, Helical wrap | Seawater | 23 | 60 | 16.26 | [96] |
| GFRP | 10, Helical wrap | Seawater | 40 | 60 | 16.84 | [96] |
| GFRP | 10, Helical wrap | Seawater | 60 | 60 | 18.17 | [96] |
| GFRP | 10, Helical wrap | Seawater | 23 | 120 | 19.9 | [96] |
| GFRP | 10, Helical wrap | Seawater | 40 | 120 | 19.63 | [96] |
| GFRP | 10, Helical wrap | Seawater | 60 | 120 | 17.15 | [96] |
| GFRP | 10, Lugs | Seawater | 23 | 60 | 18.62 | [96] |
| GFRP | 10, Lugs | Seawater | 40 | 60 | 20.71 | [96] |
| GFRP | 10, Lugs | Seawater | 60 | 60 | 20.59 | [96] |
| GFRP | 10, Lugs | Seawater | 23 | 120 | 21.2 | [96] |
| GFRP | 10, Lugs | Seawater | 40 | 120 | 19.68 | [96] |
| GFRP | 10, Lugs | Seawater | 60 | 120 | 19.93 | [96] |
| GFRP | 12, Ribbed | Seawater | 60 | 30 | 18.46 | [97] |
| GFRP | 12, Ribbed | Seawater | 60 | 60 | 18.22 | [97] |
| GFRP | 12, Ribbed | Seawater | 60 | 90 | 16.44 | [97] |
| GFRP | 12, Ribbed | Alkaline | 60 | 30 | 18.74 | [97] |
| GFRP | 12, Ribbed | Alkaline | 60 | 60 | 18.3 | [97] |
| GFRP | 12, Ribbed | Alkaline | 60 | 90 | 18.17 | [97] |
| GFRP | 12, Ribbed | Acid | 60 | 30 | 20 | [97] |
| GFRP | 12, Ribbed | Acid | 60 | 60 | 17 | [97] |
| GFRP | 12, Ribbed | Acid | 60 | 90 | 13.74 | [97] |
| BFRP | 12, Deformed surface | Alkaline | 40 | 45 | 16.48 | [98] |
| BFRP | 12, Deformed surface | Alkaline | 50 | 45 | 21.4 | [98] |
| BFRP | 12, Deformed surface | Alkaline | 60 | 45 | 20.37 | [98] |
| BFRP | 12, Deformed surface | Alkaline | 40 | 90 | 10.64 | [98] |
| BFRP | 12, Deformed surface | Alkaline | 50 | 90 | 20.59 | [98] |
| BFRP | 12, Deformed surface | Alkaline | 60 | 90 | 20.78 | [98] |
| BFRP | 12, Deformed surface | Alkaline | 40 | 180 | 15.72 | [98] |
| BFRP | 12, Deformed surface | Alkaline | 50 | 180 | 19.24 | [98] |
| BFRP | 12, Deformed surface | Alkaline | 60 | 180 | 21.81 | [98] |
| BFRP | 12, Sand-coated | Tap water | 80 | 60 | 29.4 | [97] |
| BFRP | 12, Sand-coated | Seawater | 60 | 30 | 29.4 | [97] |
| BFRP | 12, Sand-coated | Seawater | 60 | 60 | 25.6 | [97] |
| BFRP | 12, Sand-coated | Seawater | 60 | 90 | 23.9 | [97] |
| BFRP | 12, Sand-coated | Alkaline | 60 | 30 | 26.2 | [97] |
| BFRP | 12, Sand-coated | Alkaline | 60 | 60 | 26.53 | [97] |
| BFRP | 12, Sand-coated | Alkaline | 60 | 90 | 24.3 | [97] |
| BFRP | 12, Sand-coated | Acid | 60 | 30 | 23.22 | [97] |
| BFRP | 12, Sand-coated | Acid | 60 | 60 | 22.92 | [97] |
| BFRP | 12, Sand-coated | Acid | 60 | 90 | 22.74 | [97] |
| BFRP | 8, Sand-coated | Seawater | 40 | 15 | 9.54 | [99] |
| BFRP | 8, Twined | Artificial seawater | 40 | 60 | 20.8 | [100] |
| BFRP | 8, Twined | Artificial seawater | 40 | 90 | 17.8 | [100] |
| BFRP | 13, Ribbed | Artificial seawater | 50 | 270 | 8.6 | [99] |
| CFRP | 8, Ribbed | Seawater | 25 | 30 | 24.56 | [99] |
| CFRP | 8, Ribbed | Seawater | 25 | 45 | 24.01 | [99] |
| CFRP | 8, Ribbed | Seawater | 40 | 15 | 26.24 | [99] |
| CFRP | 8, Ribbed | Seawater | 40 | 30 | 28.9 | [99] |
| CFRP | 8, Ribbed | Seawater | 40 | 45 | 31.25 | [99] |
| CFRP | 8, Ribbed | Seawater | 55 | 30 | 25.64 | [99] |
| CFRP | 8, Ribbed | Seawater | 55 | 45 | 23.13 | [99] |

According to Table 4 and Figure 3, in terms of CFRP bars, the alkaline solution had a greater effect on the bond strength of these types of bars, in comparison with the seawater solution. In addition, the bond strength of BFRP bars in alkaline and seawater solutions was almost equal to approximately 20 MPa, which was lower than the bond strength of these types of bars in acid solutions. As for the GFRP bars, they were more resistant to the alkaline solutions, compared to the seawater and acid solutions. Figure 4 displays the scatter plot of the bond strengths of the different FRP bars in different environmental solutions and their trendlines.

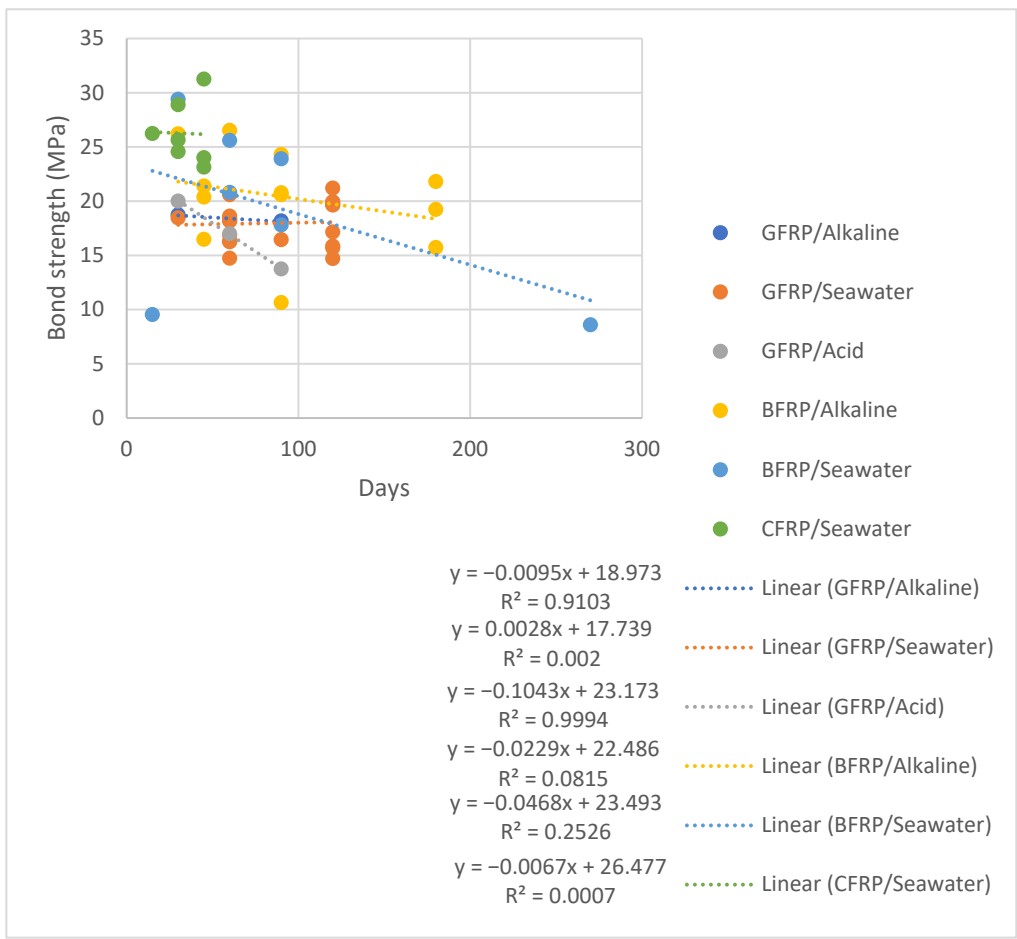

**Figure 4.** Scatter plot of bond strength of different FRP bars in different environmental solutions and their trendlines, based on data in Table 4.

Based on Figure 4, all types of FRP bars, under different environmental conditions, experienced a downward trend. The trendlines of the GFRP bars in the seawater solutions showed the lowest slope during the period. Contrary to the GFRP bars in seawater solutions, the GFRP bars in acid solutions showed the most upward trend during the period.

## 4. Influence of Hybridization on Mechanical Properties of Composite Bars

The concept of the hybridization process refers to applying different materials in one cross-section of an FRP bar to exert a pseudo-ductile behavior on fabricated hybrid FRP bars. As the different materials used for fabricating hybrid FRP bars have a different ultimate tensile strain, this leads to the ductility of fabricated FRP bars under tensile stress [44,101]. Research has shown that the use of steel materials for the fabrication of hybrid composite bars can lead to better ductility, in comparison with fabricating hybrid composite bars by only using different composite materials in one section of each hybrid composite bar. This

is due to the ductile behavior of steel fibers [102]. The application of such a ductile material for fabricating hybrid FRP bars affects the final behavior and ductility of the fabricated FRP bars.

### 4.1. Hybrid Effect on the Tensile Strength of FRP Bars

In tensile loading, FRP bars do not show plastic behavior. Composite yarns play an important role in the tensile strength of composite rebar. Wu et al. [103] found that composite yarns with different yarn-to-resin proportions have different strengths. Ma et al. [47] compared composite hybrid bars that were made of basalt fibers and steel bars that had seven glass fiber-reinforced plastic bars, to investigate their tensile strength [47]. They observed that the hybridized bars allowed a more balanced tensile behavior, which improved the ultimate tensile strength by 47%, in comparison with steel rebar. You et al. [104] tested three types of hybrid bars. E-glass fibers and carbon fibers were used to develop hybrid bars. The results indicated that the ultimate tensile strength increased by up to 5.4%. Cui and Tao [105] developed a new core-shell model of the hybrid composite bar. In this section, steel and glass fibers were randomly dispersed across the cross-section of the core, where Twaron and carbon fibers were placed within the shell. This specimen revealed a tensile strength of 628 MPa, which was 156% of that of steel bars. Seo et al. [102] developed three types of hybrid bars. The tensile performance of the recently developed hybrid bar was experimentally evaluated. The test indicated that the hybrid bar had 85% of the tensile strength of the GFRP bar. Won et al. [106] developed three types of hybrid bars, in which aramid fibers, glass fibers, and carbon fibers were utilized as yarn. Table 5 summarizes the effects of the hybridization process on the tensile behavior of FRP bars and shows the improvement of the tensile strength compared to the steel ST37 bar.

According to Table 5, the volume of the steel materials used for the fabrication of the hybrid composite bars had an adverse effect on the ultimate tensile strength of the fabricated hybrid composite bars. This was due to the lower tensile strength of the steel materials in comparison with the composite fibers.

Figure 5 demonstrates the effects of the hybridization process in improving the tensile strength, in comparison with steel bars. This was due to the use of high-tensile strength composite fibers, as opposed to steel material, for fabricating hybrid FRP bars. Considering this figure, the effectiveness of this process in increasing the tensile strength was the most evident when glass fiber and steel wire were combined. On the other hand, the combination of steel bars and basalt fibers resulted in the least improvements in terms of tensile strength.

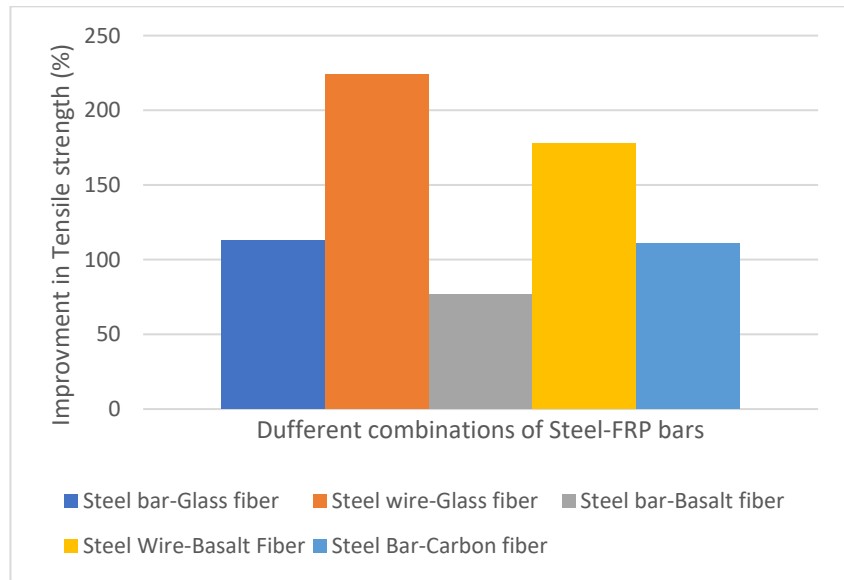

**Figure 5.** Mean value of improvement in tensile strength by hybridization process, based on data in Table 5, compared with steel bars.

**Table 5.** Effects of hybridization process on the tensile and elastic modulus of FRP bars.

| Materials | | Steel to FRP Ratio | Diameter | Tensile Strength (MPa) | Improvement in Tensile Strength | Elastic Modulus (GPa) | Reduction in Elastic Modulus | Ref. |
|---|---|---|---|---|---|---|---|---|
| Core | Crust | | | | | | | |
| Steel rod | Glass | 9.2 | 13 | 1122.7 | 203.43 | 76.5 | −61.75 | [107] |
| Steel rod | Glass | 29.2 | 13 | 1269.7 | 243.16 | 94.9 | −52.55 | [107] |
| Steel rod | Glass | 51 | 13 | 1258.8 | 240.22 | 111.2 | −44.4 | [107] |
| Steel rod | Glass | 76.2 | 13 | 833.9 | 125.38 | 148.2 | −25.9 | [107] |
| Steel wire | Glass | 9.8 | 13 | 1150.3 | 210.89 | 62.6 | −68.7 | [107] |
| Steel wire | Glass | 31.8 | 13 | 1245.4 | 236.59 | 99.8 | −50.1 | [107] |
| Steel wire | Glass | 57 | 13 | 1323.2 | 257.62 | 126.9 | −36.55 | [107] |
| Steel wire | Glass | 70.3 | 13 | 1156.4 | 212.54 | 157.3 | −21.35 | [107] |
| Steel rebar | Glass | 57.2 | 13 | 669.5 | 80.95 | 110.1 | −44.95 | [107] |
| Steel wire | Glass | 10.9 | 16 | 1232.7 | 233.16 | 58.5 | −70.75 | [107] |
| Steel wire | Glass | 36.9 | 16 | 1238.6 | 234.76 | 97.2 | −51.4 | [107] |
| Steel wire | Glass | 60.2 | 16 | 1283.1 | 246.78 | 143.3 | −28.35 | [107] |
| Steel wire | Glass | 70.1 | 16 | 1361.8 | 268.05 | 155.1 | −22.45 | [107] |
| Steel rebar | Glass | 36.6 | 16 | 779.5 | 110.68 | 100.4 | −49.8 | [107] |
| Steel rebar | GFRP | 63.2 | 16 | 596.5 | 61.22 | 146.8 | −26.6 | [107] |
| Steel–Glass | Carbon–Twaron | - | 10 | 628 | 69.73 | 142.11 | −28.945 | [105] |
| Glass | Carbon | - | 9.5 | 1191 | 221.89 | - | - | [108] |
| Steel rebar | Glass | 9.5 | 13 | 762.1 | 105.97 | 53.7 | −73.15 | [101] |
| Dispersed Steel wire | Glass | 30.8 | 13 | 688.2 | 86.00 | 98.3 | −50.85 | [101] |
| Steel rebar | Glass | 47.9 | 13 | 715.4 | 93.35 | 133.2 | −33.4 | [101] |
| Steel wire | Glass | 25 | 19 | 1217.9 | 229.16 | 90.8 | −54.6 | [102] |
| Steel wire | Glass | 42.3 | 19 | 1197.2 | 223.57 | 123.2 | −38.4 | [102] |
| Steel wire | Glass | 66.3 | 19 | 781.8 | 111.30 | 118.5 | −40.75 | [102] |
| Steel rebar | Glass | 24.7 | 19 | 899.6 | 143.14 | 88.8 | −55.6 | [102] |
| Steel rebar | Glass | 45.9 | 19 | 537.7 | 45.32 | 120.7 | −39.65 | [102] |
| Steel rebar | Glass | 67.9 | 19 | 466.6 | 26.11 | 148.2 | −25.9 | [102] |
| Carbon | Glass | - | 12.7 | 1281 | 246.22 | 80.4 | −59.8 | [104] |
| Glass | Carbon | - | 12.7 | 1083 | 192.70 | 78.9 | −60.55 | [104] |
| Dispersed Carbon | Glass | - | 12.7 | 1045 | 182.43 | 62.4 | −68.8 | [104] |
| Steel | Glass | 33.3 | 4 | 705.1 | 90.57 | 81.1 | −59.45 | [109] |
| Steel | Glass | 66.6 | 4 | 699.53 | 89.06 | 99.4 | −50.3 | [109] |
| Steel | Basalt | 66.6 | 4 | 779.66 | 110.72 | 110.4 | −44.8 | [109] |
| Steel | Basalt | 76 | 12 | 492.8 | 33.19 | 129.17 | −35.415 | [47] |
| Carbon | Aramid | - | - | 800 | 116.22 | 63 | −68.5 | [47] |
| Carbon | Glass | - | - | 550 | 48.65 | 43 | −78.5 | [47] |
| Carbon | Aramid–Glass | - | - | 503 | 35.95 | 37 | −81.5 | [47] |
| Steel bar | Basalt | 56.2 | 10 | 798.6 | 115.84 | 88 | −56 | [44] |
| Steel wire | Basalt | 28.2 | 10 | 1027 | 177.57 | 55 | −72.5 | [44] |
| Carbon | Basalt | - | 10 | 869.7 | 135.05 | 106 | −47 | [44] |
| Steel | Glass | 56.2 | 10 | 798.6 | 115.84 | 96.41 | −51.795 | [59] |
| Steel | Carbon | - | 10 | 950 | 156.76 | 129 | −35.5 | [110] |
| Steel | Carbon | - | 14 | 825 | 122.97 | 132 | −34 | [110] |
| Steel | Glass | - | 10 | 662 | 78.92 | 92 | −54 | [110] |
| Steel | Carbon | - | 12 | 716 | 93.51 | 112 | −44 | [110] |
| Steel | Carbon | - | 14 | 706 | 90.81 | 119 | −40.5 | [110] |
| Steel | Glass | - | 12 | 623 | 68.38 | 77 | −61.5 | [110] |
| Steel | Carbon | - | 16 | 700 | 89.19 | 118 | −41 | [110] |
| Steel | Basalt | - | 12.5 | 480.9 | 29.97 | 97.8 | −51.1 | [111] |
| Steel | Basalt | - | 15 | 718 | 94.05 | 108.9 | −45.55 | [112] |

*4.2. Hybrid Effect on the Modulus of Elasticity of FRP Bars*

One of the disadvantages of composite bars is their low modulus of elasticity. At high stresses, the low elasticity modulus in these bars (approximately 70GPa for GFRP, which is

approximately 33% of steel) has shown high strains too, which causes early concrete failure in the compression block of the flexural members. This section explores the effects of the hybrid process on the modulus of elasticity in composite bars.

There are three methods to measure the elastic modulus of hybrid bars. The first method uses Equation (1) to measure this property for hybrid composite bars [101], as follows:

$$E_{hybrid} = \frac{(P_1 - P_2)}{(\varepsilon_1 - \varepsilon_2)A_{Rebar}} \tag{1}$$

where $E_{hybrid}$ is the elastic modulus of the hybrid bar (Pa), $P_1$ and $P_2$ represent the applied loads at 50% and 25% of the ultimate load, respectively (N); $\varepsilon_1$ and $\varepsilon_2$. denote the strains at 50% and 25% of the ultimate load, respectively; and $A_{hybrid\ bar}$ is the cross-sectional area of the specimen (m$^2$).

The second method for measuring the elastic modulus of hybrid bars, which have been fabricated from steel and composite fibers, is by calculating the first slope of the stress–strain curve of hybrid composite bars [103].

The third method is not only utilized to estimate the tensile elastic modulus of hybrid composite bars but also to estimate the stress–strain relationship of these types of bars, which is obtained from Equation (2) [113], as follows:

$$E_{Hybrid} = \sum_i E_{11,fi}V_{fi} + E_m V_m \tag{2}$$

where $E_{11}$ is Young's modulus of fibers along the longitudinal axis; $V_{fi}$ is the volume fraction of fibers; $E_m$ is Young's modulus of the matrix; and $V_m$ is the volume fraction of the matrix.

Ma et al. [47] compared the mechanical behavior of the hybrid composite bars made from basalt fibers and steel with full GFRP bars. The results showed that the hybridization process enhanced the modulus of elasticity of the bars, in comparison with the glass fiber-reinforced plastic bars by up to 169%.

You et al. [104] tested three different cross-sections of hybrid composite bars with two full composite sections of glass or carbon fiber. Two types of saturated vinyl ester and polyester resins were made for each of the hybrid bars. The results revealed that, depending on the resin used for fabricating the hybrid FRP bars, the hybrid process increased the modulus of elasticity of the bars from 0.4% to 4.2%.

Cui et al. [105] conducted some research on hybrid bars, including steel. In these experiments, three types of fibers, namely Twaron, glass, and carbon fibers, including steel, were used to make the bars. The hybrid bars had 71% of the elasticity modulus of steel bars.

Seo et al. [101,102] conducted extensive research on comparing three types of hybrid bars with different cross-sections (a total of 140 samples) made from steel and glass fibers. The results indicated that the hybrid process could increase the modulus of elasticity of composite bars by up to 250%.

Won et al. [106] carried out extensive research on hybrid bars using carbon, aramid, and glass fibers and compared the results of the tensile test on these hybrid bars with those of the GFRP bars. The results showed an increase in the elastic modulus of all three types of composite bars, compared to GFRP bars, by up to 40%.

Hwang et al. [107] tested hybrid bars with 13 mm and 16 mm diameters, where 0.5, 1, and 2 mm steel wires were used. Note that the volume of steel wire in the sections of the hybrid bars was 10%, 30%, 50%, and 70%. One of the essential results of this paper was that increasing the steel volume in the cross-sections improved the overall elasticity modulus of the hybrid composite bars. Table 5 outlines the effects of the hybridization process on the elastic modulus of FRP bars.

Figure 6 demonstrates the effects of the hybridization process on the reduction in the elastic modulus, compared to steel bars. Considering this figure, the reduction in the elastic modulus by this process is maximized when basalt fibers and steel wire are combined.

This behavior can be attributed to the low elastic modulus of basalt fibers in comparison with other fibers. On the other hand, the combination of steel bars and carbon fibers experienced the least reduction in terms of the elastic modulus. This may be due to the high elastic modulus of carbon fibers compared to other fibers. Thus, the larger the volume of composite materials used for fabricating hybrid composite bars, the more reduction will occur in the ultimate elastic modulus of fabricated hybrid composite bars. This is due to the lower elastic modulus of composite fibers, in comparison with steel materials. Thus, the volume of the composite fibers has an adverse effect on the elastic modulus of hybrid FRP bars.

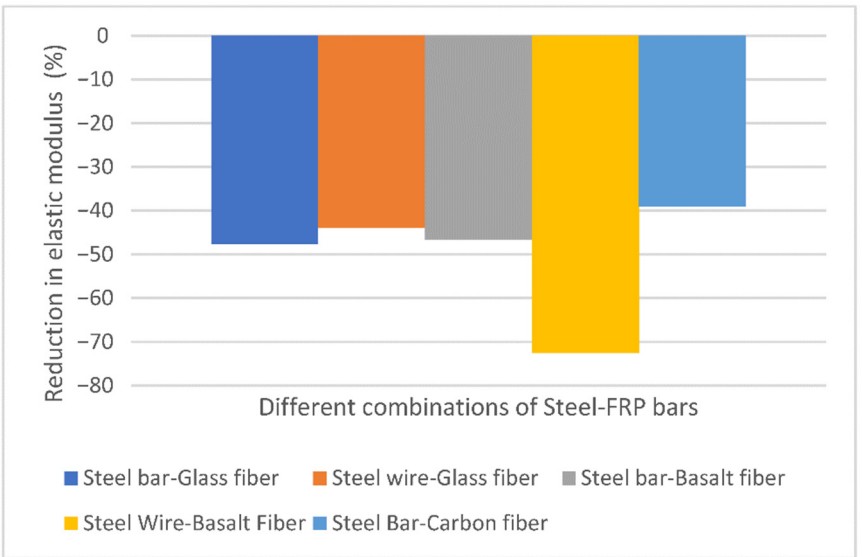

**Figure 6.** Mean value of the reduction in elastic modulus by hybridization process, based on Table 5, compared with steel bars (ST37).

*4.3. Effects of Environmental Conditions on Hybrid Bars*

As mentioned in the previous sections, one of the significant disadvantages of steel bars is their corrosion vulnerability. Corrosion in steel bars reduces the bonding of the bars to the concrete, which lowers the mechanical performance of the concrete. To resolve this problem, composite bars are being discussed. However, one of the barriers to the usage of non-steel bars in construction is their sensitive and variable behavior against environmental conditions. Previous research has shown that composites, especially GFRPs, have the potential to be degraded regarding their mechanical properties under various environmental conditions. To apply these materials as structural members, it is necessary to conduct detailed studies on their behavior and alter their mechanical properties when exposed to different environmental conditions.

Cui et al. [105] conducted a study on the corrosion of hybrid bars in an alkaline environment and compared it with composite glass bars. As noted in the previous section, one of the significant disadvantages of composite glass bars is their low resistance to alkaline environments. The results of the hybrid rebar showed that these bars could retain 93.1% of their tensile strength after eight weeks in an alkaline environment, while GFRP bars could maintain only 73.62% of their tensile strength.

Won et al. [60] performed extensive research on the durability of hybrid bars using glass, aramid, and carbon fibers in physically and chemically aggressive environmental conditions. Thus, to obtain the parameter to evaluate the durability of these bars under aggressive environmental conditions, they used tensile and interfacial shear stress (ISS) tests. The results of this paper indicate that bars made of carbon fiber and aramid have higher resistance in aggressive environments than bars made of glass fibers.

### 5. Conclusions and Research Needs

Using FRP materials as an alternative to steel bars is becoming more widespread, especially in marine and coastal areas. This is due to their acceptable mechanical behavior, such as their tensile strength, durability, and sustainability. A comprehensive review of the effects of harsh environmental conditions and hybridization processes on the tensile, bond, and elastic modulus of FRP bars have been discussed in this paper. Related conclusions and further research are needed in this area and are proposed in the following sections.

*5.1. Conclusions*

1.   In terms of tensile strength, CFRP bars are more resistant to alkaline solutions, in comparison to different FRP bars. Moreover, considering seawater and saline solutions, GFRP bars show more durability.
2.   Alkaline solutions have a greater effect on the bond strength of CFRP bars than seawater solutions. In addition, the bond strength of BFRP bars in alkaline and seawater solutions was almost equal to approximately 20 MPa, which was less than their bond strength in acid solutions. As for GFRP bars, they were more resistant to alkaline solutions than to seawater and acid solutions.
3.   When fabricating hybrid composite fibers, the type of fibers used highly affects the elastic modulus. For a high elastic modulus, carbon fibers are recommended; however, for a low elastic modulus, glass and basalt are preferred.
4.   Using the hybridization process can improve the tensile strength of fabricated hybrid FRP bars by up to approximately 210%, in comparison with steel bars (ST37). On the other hand, this process has an adverse effect on the elastic modulus of fabricated hybrid FRP bars and can reduce this mechanical behavior by up to approximately 70%, compared with steel bars' (ST37) elastic modulus.
5.   When it comes to hybrid composite bars made up of steel, the volume of the steel material has a great influence on the final mechanical behavior. The more steel used, the greater the ductility of the hybrid composite bars, as observed under tensile tests. Using steel material for fabricating hybrid composite bars generally has a positive effect on the elastic modulus of these bars. This behavior stems from the high elastic modulus of steel materials in comparison with composite fibers.
6.   Steel materials can have an adverse effect on the ultimate tensile strength of hybrid composite bars because composite fibers have a higher tensile strength than steel material.
7.   Tensile tests in the literature indicate that hybridization can improve the ductility of composite bars. Such an increase in their ductility can be attributed to using different materials in one cross-sectional area of the hybrid composite bars. Furthermore, hybridization improves the elastic modulus of composite bars and when steel is used the elastic modulus is linearly proportional to the steel's volume.
8.   Hybrid composite bars that were fabricated by steel materials show great pseudo-ductile behavior, in comparison to hybrid composite bars composed of composite fibers only. However, the former group shows lower durability compared to the latter group because of the presence of steel in their cross-sectional area.
9.   In the case of resin, epoxy and vinyl ester resins have a higher elastic modulus and tensile strength, respectively. Current data also denotes that vinyl ester and epoxy have better performance regarding their degradation level.

*5.2. Research Needs*

1.   Only a limited number of studies have investigated the effects of different environmental conditions on the compressive behavior of different FRP bars.
2.   The performance of FRP bars in cyclic loads, along with harsh environmental conditions remains obscure.
3.   The effects of the bar size and diameter on the bond and tensile behavior of FRP bars subjected to different solutions remains unclear.

4. The influence of different fibers, such as aramid fibers, on the mechanical behavior of hybrid FRP bars remains unclear.

5. The effects of the steel bars and steel wire diameter on the elastic and the tensile strength of hybrid FRP bars have not been completely investigated.

6. The effects of environmental solutions on the durability of hybrid FRP bars are still obscure.

7. The mechanical behaviors of FRP bars are deeply influenced by the types of fibers, the manufacturing process, and the types of resin, etc. These agents cause uncertainties in the final mechanical behavior of FRP bars. More research should be conducted to produce FRP bars with the same mechanical behavior. Using probabilistic models and machine learning methods can accommodate these uncertainties and provide further insight into the behavior of FRP bars that are subjected to simulated environmental conditions [114,115].

**Author Contributions:** Conceptualization, M.M. and R.H.; methodology, M.M.; software, R.H.; validation, M.M., R.H. and F.A.; formal analysis, M.M.; investigation, R.H.; resources, F.A.; data curation, S.V.A.N.K.A.; writing—original draft preparation, M.M.; writing—review and editing, R.H.; visualization, M.M.; supervision, F.A.; project administration, M.M.; funding acquisition, R.H. All authors have read and agreed to the published version of the manuscript.

**Funding:** Research received no external funding.

**Informed Consent Statement:** Not applicable.

**Conflicts of Interest:** The authors declare no conflict of interest.

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
