# Peer review of "A Comprehensive Review of the Effects of Different Simulated Environmental Conditions and Hybridization Processes on the Mechanical Behavior of Different FRP Bars"

_sustainability, doi:10.3390/su14148834_

Round 1

Reviewer 1 Report

1. The title of the paper should be double-checked and modified to exhibit the main contents of the paper. 

2. The conclusions and research needs of the paper are so general, the authors should provide a more specific comments at the end of the paper to give meaningful guidance for the possible readers. 

3. L177-188 are repeated with L174-176 in the paper. The two lines should be deleted. 

4. Mpa and Gpa in the paper are miss spelled. The letter p should be replaced with capital letter P. 

Reviewer 2 Report

Generally speaking, the paper is written well and provides the audience with a lot of information regarding the application of FRP bars. The followings are some comments that need to be addressed by the authors:

1. Line 36, the sentence is incomplete.

2. Line 287, it seems a figure or table is missing

There are also some formatting and writing issues, the authors should double-check it before re-submit the manuscript.

Reviewer 3 Report

The subject of this manuscript is well framed in the scientific domain of the Sustainability Journal. Authors conducted a review on the mechanical properties of FRP bars subjected to normal/extreme environmental conditions. The following comments must be addressed to add value to the manuscript.

General Comments:

1)  While the study results and explanation presented in this manuscript could add value in understanding the effects of different simulated environmental conditions and hybridization processes on the mechanical behavior of the investigated FRP bars, there could be more rigorous interpretation and scientific explanations of the mechanisms behind the observed trends. In the current form, the main contribution of this review is still not scientifically grown. So, it is recommended to elaborate more on your analyses.

2)  The quality of the figures is not good. This must be improved.

3)  There are a lot of grammatical and punctuation errors that must be addressed. A comprehensive English proofread is needed. Just a few examples to mention:

a.     Lines 35-36: Authors stated that “Two of the most important materials for infrastructures are concrete and [1].” Please write the other material “steel” at the end of this sentence.

b.     Lines 36-37: “Corrosion of steel bars is the main problem that causes a heavy cost for repairing or replacing concrete elements. However, this is not feasible to use steel bars as a reinforcement …” In the second sentence, “However” must be replaced with “Therefore.”

c.      In text citations are not consistent. For example, in line 111, the reference was added after the authors’ name as follows “Rifai et al. [68] tested FRP bars in alkaline solutions …”, but in lines 143-145, the reference was added at the end of the sentence “Sharaky et al. tried to assess the factors which play an essential role in the bond behavior of near-surface mounted FRP bars experimentally [73].” Please be consistent throughout the manuscript.

d.     In lines 174-176: authors stated “Figure 4 portrays the scatter plot of bond strength of different FRP bars in different environmental solutions and their trend line.” This sentence was repeated in lines 177-178: “Figure 4 portrays the scatter plot of bond strength of different FRP bars in different environmental solutions and their trendline.” Avoid redundancy!

e.      In line 237, authors used “Equation 1”, while in line 247, they used “Eq. 2.” Also, both modulus of elasticity and elastic modulus have been used interchangeably throughout the manuscript. Please be consistent!

f.       There are many other such grammatical and punctuation errors throughout the manuscript that must be addressed. For example, line 238, lines 248-249, line 278, line 287, etc.

In the Title: I suggest to remove the word “paper”, and replace “or” with “and” in the title. Also, change the term “different solutions” to a better alternative.

In Introduction; Lines 49-50: Authors stated that “One viable approach to deal with mentioned issues is using fiber-reinforced polymer (FRP) bars.” I agree. However, I believe that the authors should briefly discuss some other approaches. For example, strengthen the structural member with FRP sheet can also be mentioned. Also, using carbon nanotubes within the cement matrix may reduce the rebar corrosion. Carbon nanotubes were also found to increase the mechanical properties and other durability characteristics of concrete. The following references might be helpful to discuss these: "Performance of Composite connections strengthened with CFRP laminate", "Carbon nanotube reinforced cementitious composites: A comprehensive review", "Influence of carbon nanotubes on properties of cement mortars subjected to alkali-silica reaction", "Mechanical properties of carbon-nanotube-reinforced cementitious materials: database and statistical analysis", and "Modeling the mechanical properties of cementitious materials containing CNTs."

In Section 2.1. Composite Fibers; Lines 75-76: “Table 1 shows that the use of carbon fibers leads to higher elastic modulus and tensile strength in fabricated composite bars.” This cannot be seen in Table 1. Many other fibers possess higher tensile strength than CFRP. Please address this properly.

In Section 2.1. Composite Fibers, and Section 2.2. Resin: The discussion is too brief. Please elaborate more on the analysis and the mechanisms behind the observed trends.

In Environmental Conditions; Lines 93-94: “Combination of sodium hydroxide (NaOH), potassium hydroxide (KOH), and calcium hydroxide (Ca(OH)2) with pH values of 13.6 and 12.7 leads to simulating Alkaline solution with normal and high strength concrete environments, respectively [58,59]. To simulate ocean water, sodium chloride (NaCl) is combined with (Na2SO4) [55,60].” So, how is this related to the effect of environmental conditions on the FRP bars? This is only the simulation of various environmental conditions. There is no link between the sentences in this section. Section 3 must be rewritten. I believe Section 3 can be merged with Section 4.

In Section 4. Mechanical Properties of FRP Bars; line 101: Please add reference for ASTM D7913.

In Table 3: Please spell-out SWSSC in the note.

In Figure 1: Please have the type of FRP in the y-axis title. Also, could you add the error bars to Figures 1, 3, 5 and 6, if possible?

In Section 4; lines 122-123: “Figure 1 depicts the retention of different FRP bars in different environmental solutions.” Please mention in the text that the data used to draw this figure is based on the data presented in Table 3.

In Figures 2 and 4: It is better to include the R-squared and equations.

In Figure 3: Please have the type of FRPs in the y-axis title. Also, y-axis titles are not aligned properly. Please fix this.

In Section 5; Line 190: “…, this leads to the ductility of fabricated FRP bars under tensile strength [38,79].” Please replace “tensile strength” with either “tensile stress” or “tensile loading.”

In Figures 5 and 6: Is the steel content constant for all the composite bars shown in these figures? If not, they cannot be compared!

In Section 5.2. Hybrid effect on the modulus of elasticity; Lines 256-257: Authors stated that “The results showed that the hybrid process increased the modulus of elasticity of the bars from 0.4% to 4.2%.” Compared to steel bars? Also, in the case of using vinyl ester and polyester resins, respectively?

In Section 5.2; Lines 258-260: “Cui et al. [83] performed some research on hybrid bars, including steel. In these experiments, three types of fibers including steel were used to make bars. The hybrid bars had 71% of the elasticity modulus of steel bars.” Please mention the type of fibers used.

In Section 5.2; Line 273: Table x shows the effects of the hybridization process on the elastic 273 modulus of FRP bars.” Please replace “Table x” with “Table 5.”

In Section 6.2. Research needs; Lines 365-366: Authors stated that “These agents cause uncertainties in the final mechanical behavior of FRP bars.” As a suggestion for future research, authors may mention the development of probabilistic models to accommodate these uncertainties and provide further insight into the behavior of FRP bars subjected to various environmental conditions. The following references might be discussed to this end: "Probabilistic model for flexural strength of carbon nanotube reinforced cement-based materials", "Elastic modulus formulation of cementitious materials incorporating carbon nanotubes: Probabilistic approach", and "Probabilistic model for flexural strength of Cementitious Materials Containing CNTs."

Round 2

Reviewer 3 Report

Authors have satisfactorily addressed the comments.